# Current and Emerging Strategies for Enhancing Antibody Delivery to the Brain

**DOI:** 10.3390/pharmaceutics13122014

**Published:** 2021-11-26

**Authors:** Rinie Bajracharya, Alayna C. Caruso, Laura J. Vella, Rebecca M. Nisbet

**Affiliations:** 1Clem Jones Centre for Aging Dementia Research, Queensland Brain Institute, The University of Queensland, St Lucia, Brisbane, QLD 4072, Australia; r.bajracharya@uq.edu.au; 2The Florey Institute of Neuroscience and Mental Health, Parkville, VIC 3052, Australia; alayna.caruso@unimelb.edu.au (A.C.C.); laura.vella@florey.edu.au (L.J.V.); 3Department of Surgery, The Royal Melbourne Hospital, Australia University of Melbourne, Parkville, VIC 3052, Australia

**Keywords:** blood–brain barrier, neurological disease, antibodies, monoclonal antibodies, biotherapeutics, focused ultrasound, bi-specific antibodies, exosomes, nanoparticles

## Abstract

For the treatment of neurological diseases, achieving sufficient exposure to the brain parenchyma is a critical determinant of drug efficacy. The blood–brain barrier (BBB) functions to tightly control the passage of substances between the bloodstream and the central nervous system, and as such poses a major obstacle that must be overcome for therapeutics to enter the brain. Monoclonal antibodies have emerged as one of the best-selling treatment modalities available in the pharmaceutical market owing to their high target specificity. However, it has been estimated that only 0.1% of peripherally administered antibodies can cross the BBB, contributing to the low success rate of immunotherapy seen in clinical trials for the treatment of neurological diseases. The development of new strategies for antibody delivery across the BBB is thereby crucial to improve immunotherapeutic efficacy. Here, we discuss the current strategies that have been employed to enhance antibody delivery across the BBB. These include (i) focused ultrasound in combination with microbubbles, (ii) engineered bi-specific antibodies, and (iii) nanoparticles. Furthermore, we discuss emerging strategies such as extracellular vesicles with BBB-crossing properties and vectored antibody genes capable of being encapsulated within a BBB delivery vehicle.

## 1. Introduction

Monoclonal antibodies (mAbs) have become the predominant treatment modality for various diseases due to their high specificity and ability to be produced in large amounts at high purities [1]. Despite their success against various diseases, only a handful of mAbs have been FDA approved for the treatment of neurological diseases [2]. These include aducanumab (Aduhelm^®^) for Alzheimer’s disease, bevacizumab (Avastin^®^) for brain cancer and natalizumab (Tysabri^®^) for multiple sclerosis [2]. One of the reasons why such a small number mAbs have been approved for the treatment of neurological diseases is the inability of mAbs, and other biotherapeutics, to effectively cross the blood–brain barrier (BBB) to achieve the concentration in the brain required for efficacy. Increasing the concentration of the biotherapeutic in the brain may not only enhance antibody efficacy, leading to improved clinical trial success, but it may also reduce the cost of treatment. This is particularly important as treatment with the recently approved Aduhelm^®^ costs $56,000 (USD) per patient per year [3]. This review will therefore focus on the latest strategies for overcoming the BBB to deliver mAbs and other biotherapeutics to the brain and their application for the treatment of neurological disease. In addition, emerging strategies that show potential for achieving enhanced antibody delivery to the brain will be discussed.

### 1.1. The BBB

The BBB is a term used to describe the interface between the blood and the brain parenchyma. Its function is to tightly control the passage of molecules, ions, and cells between the blood and the central nervous system (CNS) (Figure 1) [4]. In doing so, the BBB helps to regulate brain homeostasis, mediate proper neuronal function, and protect the brain from toxins, pathogens, inflammation, injury and disease [5]. Anatomically, the BBB is comprised of highly specialized brain endothelial cells (BECs) that line the brain microvessels and are functionally supported by astrocytic end-feet processes, pericytes, and an acellular component termed the basement membrane [6]. Together, the BECs and the surrounding non-endothelial cells are referred to as the neurovascular unit [7,8].

Unlike their peripheral counterparts, BECs express tight junction proteins, such as claudins, occludins and junction adhesion molecules, and have decreased pinocytosis to restrict vesicle-mediated transcellular transport [9]. As such, these BECs link together through tight junctions to create a paracellular barrier that is highly resistant to molecules and ions, polarizing the luminal and abluminal compartments, and use efflux and nutrient transporters to control the movement of ions and molecules between the blood and the brain [9]. While this barrier function is predominantly regulated by the BECs, the surrounding non-endothelial cells of the neurovascular unit, such as pericytes and astrocytes, also play important roles. Pericytes are vascular cells embedded within the basement membrane on the abluminal surface of the BECs. They extend long cellular processes along the endothelium and facilitate cerebral blood flow regulation, angiogenesis and immune cell infiltration [10]. Astrocytes are glial cells that extend polarised cellular processes in which the end-feet of these basal processes completely ensheath the microvasculature. The astrocytic end-feet express channels, such as aquaporin-4, that help regulate the diffusion of solutes and water between blood vessels and the brain [11]. Finally, the basement membrane is an acellular component of the BBB that is comprised of a mixture of extracellular matrix proteins that are produced by the BECs, pericytes and astrocytes [12]. Specifically, the basement membrane consists of four major glycoprotein families: laminins, collagen IV isoforms, nidogens and heparan sulphate proteoglycans [12]. These extracellular proteins form a sheet that provides structural support that is important for BBB integrity [12].

### 1.2. Transport across the BBB

Under physiological conditions, molecules may cross the BBB via paracellular (between cells) or transcellular (through cells) transport [13]. As the diffusion of molecules paracellularly is limited due to the presence of the tight junctions, most molecules enter the brain transcellularly by either passive diffusion, carrier-mediated transport, receptor-mediated transcytosis (RMT), or adsorptive-mediated transcytosis [9]. Small lipophilic molecules can generally cross into the brain via passive diffusion through the BECs, while small polar molecules, such as glucose, amino acids, organic anions and cations, and nucleotides, can enter the brain by carrier-mediated transport [14]. Macromolecules like transferrin and insulin cross into the brain via RMT, while positively charged molecules like albumin enter the brain via adsorptive-mediated transcytosis [15].

### 1.3. Challenges for Antibody Delivery across the BBB

MAbs, and many other biotherapeutics, do not freely cross the BBB. This is partly due to their large size (~150 kDa). Studies using radiolabelled IgG have demonstrated that less than 0.05% of the initial dose is found in the brain parenchyma two hours after intravenous or intraperitoneal administration [16]. The mechanisms by which the small levels of IgG cross the BBB remains elusive, but the finding that the Fc fragment neonatal receptor (FcRn) is expressed on the surface of the choroid plexus and brain microvascular endothelial cells suggests that this receptor might facilitate the transit of IgG across BECs [17,18]. A recent study using an in vitro BBB derived from human-induced pluripotent stem cells, however, suggests that IgG transcytosis occurs independently of FcRn and instead originates from non-specific fluid phase endocytosis, such as macropinocytosis [19,20]. Once across the BBB, IgG is known to then undergo rapid efflux from the brain to the blood via reverse transcytosis across the BBB [21,22,23]. This has been demonstrated to be dependent on the antibody Fc domain and mediated by the FcRn [21,22,23]. Therefore, to overcome the challenges of the BBB and enhance antibody delivery to the brain, researchers have (i) engineered antibody therapeutics to create BBB-penetrating bispecific antibodies, (ii) used focused ultrasound to mechanically disrupt the BBB, or (iii) encapsulated or conjugated antibodies to nanoparticles with enhanced BBB penetrating properties (Figure 1) (Table 1). This review will focus on these current strategies, as well as emerging strategies such as exosomes and antibody gene delivery (Figure 1) (Table 1). For earlier strategies, including hyperosmotic agents and vasoactive agents, please see the review by Hersh et al. [24].

## 2. Focused Ultrasound

Focused ultrasound (FUS) is an emerging technology that facilitates the transient opening of the BBB, enabling both paracellular and transcellular transport. In this way, FUS can be used to enhance the transport of peripherally administered therapeutic molecules into the brain parenchyma. This type of BBB opening requires intravenous injection of gas-filled lipid shells, termed microbubbles (MBs), prior to the application of ultrasound. Once these MBs are in circulation and cross the field of ultrasound, they oscillate. This oscillation imparts mechanical forces upon the vasculature, causing the transient opening of tight junctions, which has been shown to increase paracellular transport [46] and vesicle-mediated transcytosis [47], allowing the extravasation of antibodies, nanoparticles and other drugs into the brain [48].

Pre-clinically, FUS-mediated BBB opening has been applied to enhance the delivery of a range of large proteins and mAbs, such as interleukin-12 (70 kDa) [49], trastuzumab [25] and bevacizumab [26], for the treatment of metastatic brain tumours and malignant glioma. Bevacizumab is an FDA-approved immunotherapy for many different types of cancers, including glioblastoma. In mice, the delivery of bevacizumab in combination with FUS was demonstrated to increase bevacizumab levels in the brain 5.7- to 56.7-fold when compared with animals that did not receive FUS. In doing so, tumour growth was supressed, and median survival time was increased by 135% compared to 48% in those animals that received bevacizumab alone [26]. Currently, a clinical trial is recruiting recurrent glioblastoma patients to test the safety and efficacy of bevacizumab in combination with FUS (clinical trial NCT04446416).

This technique has also been used to increase mAb delivery into the brain for the treatment of neurodegenerative diseases, such as Alzheimer’s disease and Parkinson’s disease [28,29,30,31,35]. Alzheimer’s disease is histopathologically characterised by the extracellular accumulation of amyloid-β (Aβ) plaques and the intracellular accumulation of pathological tau protein that deposits as neurofibrillary tangles [50,51]. Raymond et al. were the first to demonstrate the enhanced uptake of anti-Aβ antibodies into the brain parenchyma following FUS with a 2.7 ± 1.2-fold increase in anti-Aβ antibodies that were associated with plaques [28]. In a follow up study, Jordão et al. demonstrated that after four sonications in the right hemisphere with FUS, an anti-Aβ antibody could be delivered into the brain and significantly reduce plaque load [29]. A reduction in plaque load after delivering an anti-Aβ antibody with FUS has also been demonstrated in a rabbit model [30]. Recently, an antibody against a modified version of Aβ, pyroglutamate-3 Aβ, was targeted to the hippocampi of APP/PS1dE9 mice with bilateral sonication resulting in a 5.5-fold increase in mAb delivery into the brain compared to the antibody alone [31]. Here, they demonstrated that the antibody alone improved spatial learning and memory in these mice, while the combination treatment with FUS, on top of reducing hippocampal plaque load, improved these cognitive functions at a faster rate and in a higher percentage of mice [31]. Furthermore, in a transgenic mouse model of Parkinson’s disease, the FUS-mediated delivery of anti-α-synuclein mAbs enhanced the reduction in α-synuclein aggregation compared to the antibody alone [35].

### Scanning Ultrasound

In contrast to FUS, which applies single spots of sonication, an ultrasound technique that involves the application of multiple sonication spots across the entire brain, called scanning ultrasound (SUS), has been developed to increase the distribution and brain concentration of therapeutic molecules in the brain [27,32,33,34,52,53]. As opening the BBB requires the use of MBs, the process has been termed SUS^+MB^. Using this technique, the brain concentration of a tau-specific antibody in a single-chain variable fragment (scFv; 27 kDa), Fab (50 kDa) and IgG (155 kDa) format was increased by 20-, 30- and 19-fold, respectively, following intravenous administration in tau transgenic pR5 mice, demonstrating the protein size range at which SUS^+MB^ can be used to bypass the BBB [32]. In addition, weekly treatment of pR5 mice with the tau-specific scFv in combination with SUS^+MB^ revealed not only enhanced delivery to the brain (~11-fold), but also subsequent uptake into neurons, thereby enhancing tau clearance and improving associated behavioural deficits [33]. More recently, delivery of the mouse analogue of the FDA-approved anti-Aβ antibody, aducanumab (Aduhelm^®^), with SUS^+MB^ demonstrated an enhanced Aβ plaque reduction in the cortex and improved behavioural outcomes compared to single treatments alone [27]. Despite the success of SUS^+MB^ at enhancing the delivery of mAbs and other antibody fragments, a recent study combining SUS^+MB^ with a tau-specific mAb, demonstrated that the combination treatment did not reduce tau pathology any more than the mAb alone [34]. Furthermore, a correlation between increased localised antibody levels in the parenchyma and neuronal degeneration was observed. It is unclear, however, whether this effect is specific to the tau transgenic mice used or tau mAbs [34]. Therefore, whilst these studies highlight the potential of this non-invasive approach for enhanced antibody delivery to the brain, further characterization and optimisation of the delivery parameters, i.e., pressure and antibody dose, is required.

## 3. Bi-Specific Antibodies

Receptors involved in RMT, such as the transferrin receptor (TfR), insulin receptor (IR) and the low-density lipoprotein receptor-related protein-1 (LRP-1), have been hijacked for the delivery of biotherapeutics across the BBB. This process, known as the ‘molecular Trojan horse’ approach, uses a receptor-specific ligand or an antibody to ferry biotherapeutics across the BBB [54,55]. The engagement of the BBB RMT receptor on the apical surface of the BECs results in endocytosis of the conjugated biotherapeutic, allowing it to traverse across the BBB and then within the brain parenchyma to engage its therapeutic target [54,55]. This delivery approach has been reported to increase brain delivery of mAbs to 2–3% of the injected dose following intravenous administration [36,56].

### 3.1. TfR Targeting Bi-Specific Antibodies

As its name suggests, the TfR is responsible for the transcytosis of transferrin, an iron binding protein, into the brain [57]. The TfR is the most studied RMT receptor for use in the delivery of biotherapeutics to the brain. Antibody engineering has allowed a plethora of strategies utilising anti-TfR antibodies to be explored for the cross-BBB transport of antibodies [16,36,37,38,39,40,58,59,60,61]. The nature of the interaction between the bi-specific antibody and the TfR, however, seems to determine transcytosis efficiency. Reducing the affinity of the antibody to the TfR at both physiological and lysosomal pH (pH 5.5) can prevent lysosomal degradation and increase the transcytosis of the antibody into the brain parenchyma [59]. Furthermore, using an anti-TfR antibody, 8D3, with bivalent TfR binding demonstrated suboptimal transcytosis [37]. Therefore, the generation of bi-specific antibodies with monovalent binding has greatly improved transcytosis [36,37,38]. Of note, attachment of an 8D3 scFv to the C-terminal light chain ends of an anti-Aβ mAb, mAb158, resulted in monovalent binding to the TfR and increased BBB transcytosis [36]. Furthermore, attachment of an anti-TfR Fab to the C-terminal Fc domain of an anti-Aβ mAb, also resulted in monovalent TfR binding as well as the prevention of Fc-FcγR interactions and peripheral effector functions [38]. A Phase 1b/2a clinical trial using this technology has been initiated by Roche to deliver an anti-Aβ mAb (gantenerumab) for patients with mild to moderate Alzheimer’s disease (clinical trial NCT04639050). Lowering the affinity to TfR through structure-based mutations in the complementarity determining regions (CDRs) of the anti-TfR variable domains has also resulted in increased brain exposure [40]. In addition to TfR avidity, the size of the bi-specific antibody may also play a role in the effectiveness of BBB transcytosis. Fusion of the 8D3 scFv to an scFv derived from the anti-Aβ mAb, 3D6, demonstrated faster parenchymal delivery compared to the fusion of the 8D3 scFv to the IgG format of 3D6 (56 kDa versus 210 kDa, respectively) [16]. Whilst a large number of studies have utilized the anti-TfR antibody 8D3, a TfR1 VNAR fragment (single domain shark antibody), TBX2, was recently generated with similar affinity to mouse and human TfR1 [39]. Fusion of TBX2 to the anti-Aβ antibody bapineuzumab demonstrated a three-fold increase in brain concentration compared to bapineuzumab alone in both wild-type and transgenic mice overexpressing Aβ [39].

Rather than utilising an anti-TfR antibody, Denali Therapeutics Inc. engineered a small section of a human Fc domain to bind to TfR, which they termed the BBB transport vehicle (TV). This was achieved by identifying a nine-amino acid patch of the human IgG1 CH3 domain that was able to be modified in a way that did not alter the function of the Fc domain. This amino acid sequence was then randomised, resulting in a library of mutated sequences. Screening of these mutant sequences allowed for the identification of constant region clones that bound to the extracellular domain of the human TfR. These TVs can then be fused to two Fab arms with identical epitopes (mimicking a full IgG antibody) or two Fab arms with different epitopes (mimicking a bi-specific antibody) to generate antibody transport vehicles (ATVs) [41]. Using this technique, Kariolis et al. showed that an ATV against BACE1 markedly reduced brain Aβ levels compared to the parental BACE1 antibody in a model of Alzheimer’s disease as well as in non-human primates [41]. Alternatively, the TVs can also be attached to other proteins, such as enzymes, termed enzyme transport vehicles (ETVs) [62]. The generation of an ETV containing the enzyme iduronate 2-sulfatase (IDS) (ETV:IDS), a 61 kDa lysosomal enzyme that is deficient in a group of lysosomal storage diseases (specifically, Mucopolysaccharidosis Type II), demonstrated enhanced delivery to the brain and a reduction in the accumulation of toxic lysosomal lipids and proteins in a pre-clinical model of disease [62]. ETV:IDS is currently entering clinical trials (clinical trial NCT04251026), with multiple other drug targets in the pipeline using both the ATV and ETV platforms for treatment of a variety of neurological diseases.

### 3.2. IR-Targeting Bi-Specific Antibodies

The IR is expressed on the surface of BECs and is responsible for the import of insulin from the blood into the brain. Like the TfR, but to a lesser extent, the IR has also been used to facilitate the transport of biotherapeutics into the brain parenchyma [63,64]. A humanised anti-IR antibody (HIRMAb) and HIRMAb fusion proteins are currently under development by ArmaGen Technologies for the treatment of lysosomal disorders [63]. A Phase 1/2 clinical trial of HIRMab fused to a lysosomal enzyme (valanafusp alpha) demonstrated transport into both the CNS and peripheral organs due to the dual targeting of the anti-IR mAb to both the IR and the mannose-6-phosphate receptor, as well as clinical evidence of the cognitive stabilisation in paediatric patients with severe type I mucopolysaccharidosis [64] (clinical trials NCT03053089 and NCT03071341).

## 4. Nanoparticles

Nanotechnology (i.e., the study of molecules and materials with dimensions ranging from 1 to 1000 nm) has recently emerged as a promising avenue for cross-BBB drug delivery into the CNS. Nanoparticles are small, nanoscale carrier structures that harbor exciting potential for use in drug delivery and biosensing as they can be designed or modified to encapsulate or conjugate to a broad range of molecules, peptides, proteins, antibodies, and nucleic acids [65]. In addition, nanoparticles can be tailor-made in different shapes and sizes, with different levels of hydrophobicity, surface charge, and chemistry [65,66,67]. Such revisions can enable the nanoparticle to protect the drug from specific environmental and biological factors and allow for controlled drug release and site-specific targeting [45,68]. Currently, there is a wide range of nanoparticles being investigated for use in the delivery of therapeutics. Liposomes (20–1000 nm), gold nanoparticles (1–150 nm), and dendrimers (2–10 nm) are the most studied candidates that have shown promising results regarding cross-BBB drug delivery in recent years and their ability to be either conjugated or loaded with targeting or therapeutic antibodies will be discussed in more detail below. Although the majority of current studies demonstrating successful delivery of nanoparticles across the BBB encapsulate or conjugate antibodies or nanobodies for enhanced site specificity, this success encourages further study into the delivery of therapeutic antibodies into the brain onboard nanoparticle delivery complexes. It is important to note, however, that only a few nanoparticles targeting CNS diseases have ever reached the clinical trial stage, and to our knowledge all have failed due to technical and cost limitations, as well as exhibiting long-term toxicity as described in other reviews [69,70].

### 4.1. Liposomes

Artificial lipid nanoparticles or liposomes are self-assembled, spherical vesicles composed of an amphiphilic lipid bilayer or bilayers that enclose an aqueous core. This core can be modified to enable the encapsulation of different drug molecules, hydrophobic or hydrophilic, for delivery [67]. The resemblance of their lipid bilayer(s) to many biological membranes (outside of the brain) allows for effective membrane permeation and transportation of drug molecules [71]. This, along with desirable biocompatibility and biodegradability, has resulted in the great clinical establishment of liposomes for the treatment of various diseases [67]. It is important to highlight that the presence of a lipid bilayer alone is not sufficient to enable cross-BBB drug permeation [71]. RMT is the mechanism most studied for the enhancement of drug-loaded liposome cross-BBB permeation; the TfR is most targeted in existing studies. Pre-clinically, liposomes have been specifically engineered to cross the BBB via RMT for the treatment of brain cancers [42,43,72]. Liposomes loaded with the anti-cancer drug, gefitinib, and cholesterol-lowering drug, simvastatin, conjugated with T12, a peptide specific to TfR, and a nanobody specific to a tumour-associated macrophage marker, PD-L1, have been demonstrated to engage the TfR, cross the BBB, and then target tumour-associated macrophages in mice with an intracranial transplant of non-small lung cell cancer cells [72]. This cross-BBB mechanism has also been exploited using H-ferritin liposomes that harbour intrinsic TfR selectivity to deliver therapeutic mAbs, cetuximab or trastuzumab, and target cancer cells within the CNS [42]. These liposome drug complexes were demonstrated to cross a polarized human microvascular endothelial cell monolayer (hCMEC/D3), a common in vitro BBB model, whilst maintaining their functionality [42]. Although the TfR is prominently established as a capable target for brain drug delivery, studies have shown that other RMT receptor candidates display more favourable expression patterns for this function. LRP1 demonstrates a high degree of expression in the brain endothelium and low expression in the parenchyma and peripheral tissues [73]. These parameters are highly desirable for the delivery of therapeutics into the CNS. A liposome conjugated with angiopep-2, a peptide specific to BBB-localised LRP1, and CD133 mAbs, for targeting glioblastoma stem cells (GSCs), has been designed to encapsulate and enhance the site-specific delivery and therapeutic efficacy of the existing chemotherapy drug, temozolomide (TMZ) [43]. This dual-targeting liposome demonstrated enhanced in vitro BBB permeability, and in vivo demonstrated heightened survival time, enhanced anti-tumour effects, site-specific targeting, and negligible toxicity in GSC-bearing mice [43].

### 4.2. Metallic Nanoparticles

Metallic nanoparticles are generally much smaller than the other classes of nanoparticles, e.g., liposomes, and can be comprised of pure metal particles such as gold, silver, and iron, alongside metallic allotropes of non-metals. They are dense, spherical carrier structures that, due to their solid composition, do not encapsulate drug molecules for delivery [66]. Alternatively, therapeutic molecules can be conjugated to the external surface of metallic nanoparticles [66]. These nanoparticles have been shown to cross the BBB through passive transmembrane diffusion and RMT. Additionally, some studies have explored the use of the olfactory nerve for direct delivery into the brain; however, they observed very low levels of drug absorption [65,68,74]. Gold nanoparticles (AuNPs) conjugated with monovalent antibodies targeting TfR and BACE1 have been shown to successfully cross both in vitro and in vivo BBB models via TfR-mediated transcytosis [44]. The use of monovalent antibodies was shown to be superior to the use of bivalent antibodies specific to TfR, especially when used at low functionalization (<10 antibodies per AuNP), exhibiting a three-fold increase in brain parenchymal uptake [44]. In addition to exhibiting AuNP transport across the BBB, the authors note that BACE1 is a potential therapeutic target for Alzheimer’s disease. Additionally, although they were unable to fully explore the therapeutic capabilities of their AuNP formulations, this is an area of study that should be explored in future work.

### 4.3. Dendrimers

A dendrimer is a unique type of polymeric nanoparticle that conforms to a radially symmetric and highly organized structure consisting of tree-like branched monomers dispersing from a central core. Dendrimers are typically much smaller than liposomes, with some AuNPs only reaching up to 10 nm in width, and they therefore rarely encapsulate drug molecules as liposomes do. Instead, therapeutics can be conjugated to their branched surfaces for delivery [69]. Cross-BBB delivery of dendrimers has been achieved through carotid injection [75], olfactory nerve transport [76], and less invasively via exploitation of carrier-mediated transport and endocytosis mechanisms [77]. Polyamidoamine (PAMAM) is one of the more established dendrimers currently being studied for use in cross-BBB drug delivery, recently demonstrating very promising in vitro and in vivo results [45,69]. A PAMAM dendrimer loaded with siLSNCT5, a type of non-coding RNA that contributes to the suppression of tumour growth and glioma metastasis, was conjugated to cell-penetrating peptide, tLyp-1, and administered intravenously in glioma-bearing mice. In addition, the dendrimer was conjugated to an anti-NKG2a mAb to promote anti-tumour immunity via T and NK cell activation [45]. This in vivo model demonstrated cross-BBB uptake via caveolae-mediated endocytosis and tumour-specific targeting followed by reduced glioma cell growth and migration and increased survival time [45].

## 5. Exosomes: An Emerging Strategy

One of the most prominent examples of natural carriers is a class of small, membrane bound vesicles known as exosomes. Exosomes are secreted by all cell types and tissues and act as messengers within the body. They are released from a ‘donor cell’ carrying biologically active molecules, such as protein and nucleic acid, and they traffic these molecules to local or distant ‘recipient cells’ to deliver their cargo and alter the function of the recipient cell [78,79,80]. Exosomes have several features that make them a promising drug delivery system, including a lipid bilayer that reduces renal clearance and serves to encapsulate and protect cargo that would otherwise be degraded, homing properties that enable recipient cells to recognize and take up the vesicles, and low immunogenicity [81]. The wide diversity of endogenous marker molecules present on the exosome surface and their low immunogenicity make them advantageous for drug delivery when compared to lipid-based nanoparticles. One of the most relevant key features of exosomes, however, is their ability to be ‘loaded’ with a therapeutic molecule that, depending on the cell type they have been derived from, can cross the BBB [82,83,84,85].

Recently, exosomes derived from blood have been loaded with either flavonoid, quercetin [82], or dopamine [83], and demonstrated to cross the BBB in mice following intravenous injection. Although the cargo molecules above were not antibodies, these studies demonstrate the ability of exosomes to be loaded post-isolation and their ability to cross the BBB in vivo. Additionally, exosomes isolated from neural stem cells (NSCs) have been engineered to package the fluorescent protein mCherry [84]. This was achieved by stably expressing mCherry fused to an XP tag, a tag that directs the protein to the inner surface of the exosome membrane. The mCherry loaded exosomes were demonstrated to cross an in vitro BBB without hampering the endothelial cell monolayer integrity [84]. Transcytosis occurred through the interaction of the NSC exosomes with BECs through heparan sulfate proteoglycans, resulting in the dynamin-dependent endocytosis of exosomes into these cells [84]. The loading of exosomes with exogenous cargo can also be achieved by exploitation of an endogenous late-domain loading pathway. This mechanism was used to engineer exosomes containing Cre recombinase that were directed towards the neurons of mice following intranasal delivery [85]. Cre recombinase and mCherry are 34 kDa and 27 kDa in size, respectively, suggesting that this technology could be used to specifically package therapeutic antibody fragments or nanobodies into exosomes. Exosomes derived from other cell types such as dendritic cells [86,87], BECs [88], macrophages [89,90], and mesenchymal stromal cells [91] have been demonstrated to cross the BBB in vitro and in vivo [92]. This is in the absence of a protein cargo, however.

## 6. Vectored Antibody Genes: The Future of Antibody Delivery?

The high cost associated with recombinant antibody production and the difficultly of transporting large proteins across the BBB suggests that vectored antibody gene delivery may be a cheaper and easier approach. Adeno-associated virus (AAV), exosomes, and other non-viral vectors are increasingly being utilised for the delivery of DNA across the BBB and into cells [93,94,95,96]. Of note, the AAV9-based onasemnogene abeparvovec-xioi (Zolgensma^®^) was approved by the US Food and Drug Administration for infantile spinal muscular atrophy in patients younger than two years of age [97]. Onasemnogene abeparvovec-xioi is a recombinant, AAV9 vector-based gene therapy that delivers a transgene encoding the human survival motor neuron (SMN) protein driven by the cytomegalovirus enhancer/chicken-β-actin (CBA, also known as the CAG promoter) hybrid promoter. Advantageously, this therapy is administered just once intravenously and results in long-lasting expression of SMN [98,99]. Although SMN is not an antibody, it is roughly the same size as an scFv.

Since discovering that the AAV9 serotype can cross the BBB with some success, the generation of AAV9 variants with enhanced BBB penetration have been developed, including AAV-PHP.B, which increases the transduction of cells in the CNS 40-fold [100]. This serotype was further optimised a year later with the generation of another variant, AAV-PHP.eB, which has 2.5-fold greater transduction per cell throughout the cortex and hippocampus and transduces a greater percentage of cortical and striatal cells than AAV-PHP.B [101]. However, it was recently demonstrated that the increased BBB permeability of these variants of AAV9 require the presence of the brain microvascular endothelial cell protein, LY6A (SCA-1), which is not expressed in humans, nor in some mouse strains (such as, BALB/c- and BALB/c-related mouse lineages) [102]. An alternate AAV9 variant, termed AAV-F, is able to transduce both C57BL/6 and BALB/c strains [103], providing an advantage for pre-clinical mouse studies but not for translation into humans.

Although they are a promising strategy for enhancing therapeutic antibody concentrations in the brain, AAV vectors have several limitations, one of which is their small packaging capacity (<5 kb) [104]. However, as an IgG heavy and light chain are within this size capacity, this should not be a limitation of vectored mAbs. This is evident in recent studies in which AAV vectors have been used to administer an anti-tau mAb directly into the hippocampus of tau transgenic mice, resulting in high hippocampal antibody levels and reduced tau pathology [105,106]. Despite not crossing the BBB, these studies show the ability of neurons to express a complete and functional IgG and smaller antibody fragments in the brain. Furthermore, one of the limitations of delivering AAV into humans is the presence of AAV neutralizing antibodies that have arisen as a result of previous exposure to the wild-type virus [107]. It is estimated that between 20–40% of the population have neutralizing antibodies against any given AAV serotype at a titre that can prevent successful transduction [108]. Additionally, high doses of AAV can cause liver toxicity [109]. A potential way to overcome these limitations is to use exosome-associated AAVs (vexosomes) that have been demonstrated to evade neutralizing antibodies as well as have greater transduction efficiency compared to the standard AAV [110]. Alternatively, the use of non-viral brain delivery vectors for gene therapy is also under active investigation. Recently PEGylated liposomes were conjugated to a TfR monoclonal antibody and used to encapsulate the gene encoding human Niemann–Pick C1 (NPC1) protein to treat the lysosomal cholesterol storage disorder Niemann–Pick C1 disease. Treatment of NPC1 null mice resulted in NPC1 mRNA expression in the brain, spleen and liver and a reduction in tissue inclusion bodies in the brain [96]. This study used the neuronal-specific human platelet derived growth factor-B (PDGFB) promoter to drive the expression of NPC1, which demonstrated a high degree of transgene expression in neurons [96]. This raises an important point because whilst the commonly used CBA promoter achieves brain wide transduction, it also drives expression in other tissues outside of the brain [111]. Therefore, a neuronal specific promoter, such as synapsin or PDGFB, may be a better alternative for brain expression [111].

## 7. Conclusions

Enhancing the delivery of antibodies and other biotherapeutics across the BBB for the treatment of neurological diseases will not only improve treatment outcomes but also reduce production and treatment costs and the number of treatments required. Excitingly, there are now several clinical trials underway to determine the safety and efficacy of RMT shuttles (Table 2). Whilst the TfR-targeting RMT has dominated the field, it is important to consider other RMT receptors that may have better expression at the BBB compared to peripheral tissues within humans. Here, further exploration into BEC transporters, their relative abundance and their mechanisms of cross-BBB transport will be required. Furthermore, the majority of studies to date have delivered full-sized mAbs (Table 1). Whilst these studies have been successful, smaller antibody fragments may be an attractive alternative to full-sized antibodies, particularly when size restrictions apply. Lastly, the combination of different strategies, such as focused ultrasound delivery of nanoparticles, may overcome the limitations of individual strategies to enhance BBB delivery.

## Figures and Tables

**Figure 1 pharmaceutics-13-02014-f001:**
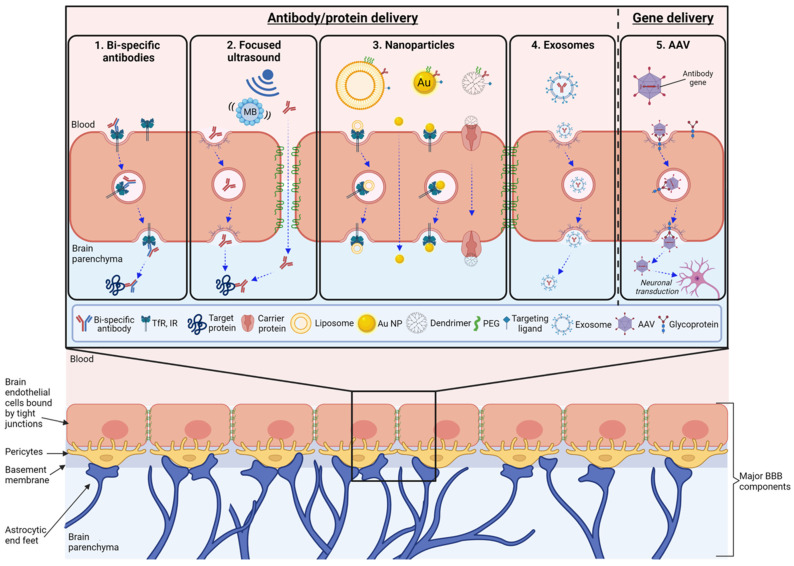
Current and emerging delivery strategies to enhance antibody penetration in the brain. The blood–brain barrier (BBB), composed of endothelial cells (BECs), the acellular basement membrane, pericytes and astrocytic end-feet, shields the brain microenvironment from foreign materials in the blood, including administered biotherapeutics. Strategies to overcome the BBB and enhance the delivery of biotherapeutics, particularly mAbs, include: (1) Bi-specific antibodies. Conjugation of a therapeutic antibody to an antibody or ligand specific to an RMT receptor (e.g., transferrin receptor (TfR) or insulin receptor (IR)) results in endocytosis, allowing the antibody to traverse across the BBB and then engage its CNS target within the brain parenchyma. (2) Focused ultrasound in combination with the intravenous injection of gas-filled lipid shell microbubbles (MBs) imparts a mechanical force upon the brain endothelium, inducing increased vesicle formation within BECs and disruption of tight junctions between BECs, allowing peripherally administered antibodies to cross the BBB through vesicle-mediated transcytosis and paracellular transport, respectively. (3) Nanoparticles describe a range of nanoscale drug delivery vehicles including liposomes (20–1000 nm), gold nanoparticles (Au NPs) (1–150 nm) and dendrimers (2–10 nm). Antibodies can be conjugated to the outer surface of the nanoparticle. In addition, nanoparticles can be conjugated with targeting ligands and polyethylene glycol (PEG) to mediate efficient transcytosis across the BBB and site-specific targeting. These nanoparticles have been demonstrated to cross the BBB via receptor-mediated transcytosis, transmembrane diffusion, and carrier-mediated transport. (4) Exosomes can be loaded with proteins and have been shown to cross the BBB through its interaction with heparan sulfate proteoglycans on the surface of BECs, resulting in endocytosis and subsequent transcytosis across the BBB. This is an emerging strategy and could be used to encapsulate therapeutic antibody fragments or nanobodies in the future; (5) Adeno-associated virus (AAV). Viral vectors encoding protein/antibody genes and packaged into BBB-crossing AAV serotypes likely cross the BBB through glycoprotein-mediated transcytosis (akin to RMT), subsequently resulting in neuronal transduction and expression of protein/antibody genes. This approach would be especially useful for intracellular protein targets. Created with BioRender.com.

**Table 1 pharmaceutics-13-02014-t001:** Strategies that have enhanced therapeutic antibody delivery to the brain.

Delivery Strategy	Therapeutic Antibody (Size)	Antibody Target	Disease Model	Method of Transport	Reference
Focused ultrasound (FUS; SUS; SUS^+MB^)	Trastuzumab/Herceptin^®^ (148 kDa)	Human epidermal growth factor receptor (HER2)	Breast cancer with brain metastasis	N.D.	[25]
	Bevacizumab/Avastin^®^ (149 kDa)	Vascular endothelial growth factor (VEGF)	Malignant glioma	N.D.	[26]
	Aducanumab/Aduhelm^®^ (146 kDa)	Oligomeric Amyloid-β peptide (Aβ)	Alzheimer’s disease	N.D.	[27]
	Anti-Aβ polyclonal (Rabbit sera) (~150 kDa)	Amyloid-β peptide (Aβ)	Alzheimer’s disease	N.D.	[28]
	Anti-Ab mAb, (100–150 kDa)	Amyloid-β peptide (Aβ)	Alzheimer’s disease	N.D.	[29,30]
	Anti-Aβ mAb, A07/2a (100 kDa)	pGlu3 Amyloid-β peptide (Aβ)	Alzheimer’s disease	N.D.	[31]
	Anti-2N tau antibody, RN2N scFv (27 kDa) Fab (56 kDa) IgG (155 kDa)	Tau (isoforms with two N-terminal domains)	Fronto-temporal lobar degeneration	N.D.	[32,33]
	Anti-tau mAb (RNF5) (150 kDa)	Tau (all isoforms)	Fronto-temporal lobar degeneration	N.D.	[34]
	Anti-α-synuclein mAb (150 kDa)	α-synuclein	Parkinson’s disease	N.D.	[35]
Bi-specific antibody	Anti-Aβ mAb, (150 kDa)	Amyloid-β peptide (Aβ)	Alzheimer’s disease	RMT (TfR)	[36,37,38,39]
	Anti-Aβ mAb, 13C3 (150 kDa)	Protofibrillar Amyloid-β peptide (Aβ)	Alzheimer’s disease	RMT (TfR)	[40]
	Anti-Aβ antibody, 3D6 scFv (~30 kDa) IgG (~150 kDa)	Amyloid-β peptide (Aβ)	Alzheimer’s disease	RMT (TfR)	[16]
	Anti-BACE1 mAb (~150 kDa)	BACE1	Alzheimer’s disease	RMT (TfR)	[41]
Nanoparticle (liposome)	Cetuximab (152 kDa)	Epidermal growth factor receptor (EGFR)	In vitro BBB model (brain malignancies)	RMT (TfR)	[42]
	Trastuzumab (148 kDa)	HER2 receptor	In vitro BBB model (brain malignancies)	RMT (TfR)	[42]
	Anti-CD133 mAb (~150 kDa)	Glioblastoma stem cells (GSCs)	Glioblastoma	RMT (LRP-1)	[43]
Nanoparticle (gold)	Anti-BACE1 mAb (~150 kDa)	BACE1	Alzheimer’s disease	RMT (TfR)	[44]
Nanoparticle (dendrimer)	Anti-NKG2a mAb (~150 kDa)	NK cell receptor	Glioblastoma	RMT (caveolae-mediated endocytosis)	[45]

Abbreviations: N.D., not determined; RMT, receptor-mediated transport; TfR, transferrin receptor; LRP-1, lipoprotein receptor-related protein-1.

**Table 2 pharmaceutics-13-02014-t002:** Current delivery strategies in clinical trials.

Delivery Strategy	Drug Name	Disease	Company	Clinical Trial Phase	Status	ClinicalTrials.gov ID
Focused ultrasound	Device: NaviFUS Drug: Bevacizumab (Avastin^®^)	Recurrent glioblastoma	NaviFUS Corporation	N/A	Recruiting	NCT04446416
Bi-specific antibody (TfR)	RO7126209 (Brain shuttle gantenerumab)	Mild to moderate Alzheimer’s disease	Hoffmann-La Roche	Phase 1b/2a	Recruiting	NCT04639050
Bi-specific antibody (TfR)	DNL310 (Enzyme transport vehicle-Iduronate-2-sulfatase fusion, ETV:IDS)	Mucopolysaccharidosis Type II (Hunter syndrome)	Denali Therapeutics Inc.	Phase 1/2	Recruiting	NCT04251026
Bi-specific antibody (HIR)	AGT-181, valanafusp alpha (HIRMab-Human alpha-L-iduronidase fusion)	Mucopolysaccharidosis Type I	ArmaGen, Inc.	Phase 1/2 Phase 1/2	Completed Completed	NCT03053089 NCT03071341

N/A = not applicable; TfR = transferrin receptor; HIR = human insulin receptor.

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
