# Peer review of "Current and Emerging Strategies for Enhancing Antibody Delivery to the Brain"

_pharmaceutics, 2021, doi:10.3390/pharmaceutics13122014_

Round 1
Reviewer 1 Report
In this manuscript, the authors have described the delivery methods of antibody to the brain. This manuscript is well-described, and this review provides an important contribution to this field. I have some concerns about this manuscript as described below.
Comments
- The reviewer is wondering that what is the best %ID of antibody, and what method is the best so far.
- The reviewer suggests that the addition of the transport mechanism of IgG at the BBB, and the explanation why antibodies are unable to cross the BBB.
- Exosome is prominent strategy, but there is no evidence. As this manuscript is a review, I do not need the section related to exosome or add this section to nanoparticles.
Author Response
We thank the reviewer for their time as well as their valuable comments and suggestions. We believe this feedback has greatly improved our manuscript. Please find our response to the individual comments below:
Reviewer 1:
1. The reviewer is wondering that what is the best %ID of antibody, and what method is the best so far.
Thank you for this suggestion. We have now included the fold change in brain delivery or the %ID of antibody where this information was available. Because these results vary between the experimental conditions used (for example the mouse used and differences in WT mice versus transgenic mice) we have avoided speculating which technique is best.
2. The reviewer suggests that the addition of the transport mechanism of IgG at the BBB, and the explanation why antibodies are unable to cross the BBB.
We have now included a paragraph ‘Challenges for drug delivery across the blood-brain barrier’ that provides an explanation as to why antibodies are unable to cross the BBB naturally and the proposed mechanisms by which a limited concentration of antibody is able to cross into the brain.
3. Exosome is prominent strategy, but there is no evidence.As this manuscript is a review, I do not need the section related to exosome or add this section to nanoparticles.
We thank the reviewer for this suggestion. We have now clarified in the manuscript that exosomes are an emerging technique that holds promise for enhanced delivery of antibodies to the brain. Specifically, we have highlighted exosomes as an emerging technique in the abstract and have changed the title of the section 5 to ‘Exosomes: an emerging strategy’.
Reviewer 2 Report
This review manuscript discussed the strategies for enhancing delivery of antibodies through the blood-brain barrier. Overall, this manuscript has a lot of qualities, it is well designed, the text is comprehensive and generally well organized into sections. New and significant findings have been highlighted in the manuscript.
There are few concerns/suggestions for the manuscript modification.
The abstract should be more informative, it should contain at least the list of current and emerging strategies for enhancing antibody delivery to the CNS.
One of the major concerns is the absence of references after some parts of the text, even the whole paragraphs. Some of the examples are: the whole first section (1. Introduction), line 47, lines 350-363, etc. The whole manuscript should be carefully checked by the authors for missing references. On the other hand, there are references in the ‘Conclusion’ section, which in my opinion should be in the other parts of the manuscript.
International nonproprietary names (INNs) as official generic names given to pharmaceutical active ingredients should be written with initial lowercase letters (trastuzumab, bevacizumab…). The names of proprietary products are distinguished by initial capital letters and the registered trademark symbol; for example, bevacizumab (Avastin®) instead of Bevacizumab (Avastin).
The title of the section 1.2. (‘Transport across the BBB’) is stated twice in the manuscript (line 69 and line 81). Besides, some parts of the text are similar in these two paragraphs:
‘Small lipophilic molecules can generally cross into the brain via passive diffusion through the BECs’ (line 75) and transmembrane diffusion ‘is favored by molecules of a low molecular weight and high degree of lipid solubility’ (line 84).
The sentence ‘researchers have combined the delivery of biotherapeutics with strategies to enhance their delivery across the BBB’ (lines 95-96) should be rephrased and clarified.
The creation of the new table containing the information on the current clinical trials based on discussed strategies would be beneficial.
Line 291: simvastatin is not an anticancer drug
The use of abbreviations should be also checked by the authors. For example, ‘receptor-mediated transcytosis’ (line 319) has already been defined as RMT in the text.
Line 386: ‘vectored antibody delivery’ – it should be clarified that it is not actually delivery of antibodies but of the genes.
The sentence ‘patients must have a low anti-AAV antibody titre prior to treatment to ensure safety’ (line 419) should be clarified that neutralizing antibodies to AAV can be found in humans and some animal species as a result of exposure to the wild-type virus and that it mainly affects the efficacy of the treatment (more than safety).
At the end of section 6, the strategies for gene therapy with non-viral vectors should be also mentioned. For example, Jiang et al. (2020) designed transferrin receptor targeting monoclonal antibody conjugated PEGylated liposomes for gene therapy (via plasmid DNA) of a lysosomal cholesterol storage disorder, Niemann-Pick C1, which affects brain severely (ref: Jiang D, Lee H, Pardridge WM. Plasmid DNA gene therapy of the Niemann-Pick C1 mouse with transferrin receptor-targeted Trojan horse liposomes. Scientific Reports. 2020;10:13334).
The conclusion should be the section 7 and not 6.
Author Response
We thank the reviewer very much for their time as well as their valuable comments and suggestions. We believe this feedback has greatly improved our manuscript. Please find our response to the individual comments below:
Reviewer 2:
1. The abstract should be more informative, it should contain at least the list of current and emerging strategies for enhancing antibody delivery to the CNS.
In the revised manuscript, the abstract has been edited to be more informative and separately lists current and emerging strategies for enhancing antibody delivery to the brain.
2. One of the major concerns is the absence of references after some parts of the text, even the whole paragraphs. Some of the examples are: the whole first section (1. Introduction), line 47, lines 350-363, etc. The whole manuscript should be carefully checked by the authors for missing references. On the other hand, there are references in the ‘Conclusion’ section, which in my opinion should be in the other parts of the manuscript.
We apologise for this oversight. The manuscript has been thoroughly read and the relevant references have been inserted. We have also removed references from the conclusion.
3. International nonproprietary names (INNs) as official generic names given to pharmaceutical active ingredients should be written with initial lowercase letters (trastuzumab, bevacizumab…). The names of proprietary products are distinguished by initial capital letters and the registered trademark symbol; for example, bevacizumab (Avastin®) instead of Bevacizumab (Avastin).
In the revised manuscript, international non-proprietary names are written with an initial lowercase letter and the propriety names have been written with a capital letter with the registered trademark symbol.
4. The title of the section 1.2. (‘Transport across the BBB’) is stated twice in the manuscript (line 69 and line 81). Besides, some parts of the text are similar in these two paragraphs:
‘Small lipophilic molecules can generally cross into the brain via passive diffusion through the BECs’ (line 75) and transmembrane diffusion ‘is favored by molecules of a low molecular weight and high degree of lipid solubility’ (line 84).
In the revised review we have clarified this and created two distinct sections: Section 1.2 ‘Transport across the BBB’ and section 1.3 ‘Challenges for antibody delivery across the BBB’. Section 1.2 describes physiological transport across the BBB, whereas section 1.3 describes the challenges faced by antibodies, specifically.
5. The sentence ‘researchers have combined the delivery of biotherapeutics with strategies to enhance their delivery across the BBB’ (lines 95-96) should be rephrased and clarified.
We have now rephrased and expanded this sentence for clarity. It now reads ‘Therefore, to overcome the challenges of the BBB and enhance antibody delivery to the brain, researchers have (i) engineered antibody therapeutics to create BBB-penetrating bispecific antibodies, (ii) used focused ultrasound to mechanically disrupt the BBB, or (iii) encapsulated or conjugated antibodies to nanoparticles with enhanced BBB penetrating properties.’ (Lines 101- 108)
6. The creation of the new table containing the information on the current clinical trials based on discussed strategies would be beneficial.
We thank the reviewer for the great suggestion and have included a new table as Table 2.
7. Line 291: simvastatin is not an anticancer drug
Thank you for alerting us to this error. We have now removed ‘anticancer’ from the text and replaced it with ‘cholesterol-lowering’ (Line 316)
8. The use of abbreviations should be also checked by the authors. For example, ‘receptor-mediated transcytosis’ (line 319) has already been defined as RMT in the text.
We have thoroughly read the manuscript to ensure the correct use of abbreviations.
9. Line 386: ‘vectored antibody delivery’ – it should be clarified that it is not actually delivery of antibodies but of the genes.
We have now edited the manuscript to read ‘Vectored antibody gene deliver’ to clarify that antibody genes have been delivered.
10. The sentence ‘patients must have a low anti-AAV antibody titre prior to treatment to ensure safety’ (line 419) should be clarified that neutralizing antibodies to AAV can be found in humans and some animal species as a result of exposure to the wild-type virus and that it mainly affects the efficacy of the treatment (more than safety).
This sentence has been clarified to read ‘Furthermore, one of the limitations of delivering AAV into humans is the presence of AAV neutralizing antibodies that have arisen from the result of previous exposure to the wild-type virus [107]. It is estimated that between 20-40% of the population have neutralizing antibodies against any given AAV serotype at a titre that can prevent successful transduction [108].’ (Lines 446-450)
11. At the end of section 6, the strategies for gene therapy with non-viral vectors should be also mentioned. For example, Jiang et al. (2020) designed transferrin receptor targeting monoclonal antibody conjugated PEGylated liposomes for gene therapy (via plasmid DNA) of a lysosomal cholesterol storage disorder, Niemann-Pick C1, which affects brain severely (ref: Jiang D, Lee H, Pardridge WM. Plasmid DNA gene therapy of the Niemann-Pick C1 mouse with transferrin receptor-targeted Trojan horse liposomes. Scientific Reports. 2020;10:13334).
Thank you for this suggestion. We have now mentioned this study in ‘Section 6: Vectored antibody gene delivery: the future of antibody delivery:’ (Lines 454-461)
12. The conclusion should be the section 7 and not 6.
We apologize for this oversight. The conclusion is now listed as section 7.
Reviewer 3 Report
A fine review.
Author Response
We thank the reviewer for their time and valuable comments. Please find our response to the individual comments below:
Reviewer 3:
- A fine review
Thank you.
Round 2
Reviewer 2 Report
In my opinion, the manuscript is much improved and I support it to be accepted in this form.